# The Role of Sound in Livestock Farming—Selected Aspects

**DOI:** 10.3390/ani13142307

**Published:** 2023-07-14

**Authors:** Katarzyna Olczak, Weronika Penar, Jacek Nowicki, Angelika Magiera, Czesław Klocek

**Affiliations:** 1Department of Horse Breeding, National Research Institute of Animal Production, Krakowska St. 1, 32-083 Balice, Poland; 2Department of Animal Genetics, Breeding and Ethology, Faculty of Animal Sciences, University of Agriculture in Kraków, 24/28 Mickiewicza Ave., 30-059 Cracow, Poland; weronika.penar@gmail.com (W.P.); jacek.nowicki@urk.edu.pl (J.N.); angelika.magiera0@gmail.com (A.M.); rzklocek@cyf-kr.edu.pl (C.K.)

**Keywords:** sound, livestock, animal welfare, living conditions, arousal, farms

## Abstract

**Simple Summary:**

Greater awareness about animal welfare forces breeders to pay more attention to animals’ needs and behaviors. The presence of sound coming from surroundings, machines, people, and animals themselves may be an important factor that affects animal welfare. If understood correctly, vocalization can provide valuable information about the emotional state of an animal, thus allowing situations to be adjusted to keep them comfortable. On the other hand, excessive noise can significantly affect the health and behavior of farm animals. If the sounds are outside the hearing range of humans, it becomes challenging to recognize their influence unless one has a deep understanding of how farm animals’ hearing works. In summary, existing research provides promising insights into the connection between sounds and the welfare of farm livestock. While there are reports highlighting the harmful effects of noise on livestock, further research should focus on exploring this aspect in more detail.

**Abstract:**

To ensure the optimal living conditions of farm animals, it is essential to understand how their senses work and the way in which they perceive their environment. Most animals have a different hearing range compared to humans; thus, some aversive sounds may go unnoticed by caretakers. The auditory pathways may act through the nervous system on the cardiovascular, gastrointestinal, endocrine, and immune systems. Therefore, noise may lead to behavioral activation (arousal), pain, and sleep disorders. Sounds on farms may be produced by machines, humans, or animals themselves. It is worth noting that vocalization may be very informative to the breeder as it is an expression of an emotional state. This information can be highly beneficial in maintaining a high level of livestock welfare. Moreover, understanding learning theory, conditioning, and the potential benefits of certain sounds can guide the deliberate use of techniques in farm management to reduce the aversiveness of certain events.

## 1. Introduction

Modern society is showing increasing concern for animal welfare. The term social license to operate (SLO) refers to society’s trust in a company, organization, or entity operating within legal, cultural, social, and ethical norms, and is more and more often applied to animal protection [1].

Broom (1986) defined welfare as an animal’s physical and mental state in relation to its attempts to cope with its environment; thus, the focus was to minimize the negative experiences [2]. Nowadays, more attention is being paid to positive emotions, which are the key to a good animal life [3]. Taking into account social license to operate and increased understanding of animal behavior, the current model of the five freedoms falls short of adequately assessing animal welfare. The five-domain model has been developed to encompass additional aspects of mental well-being. In other words, animals should have a “life worth living” [4].

To ensure living conditions are suitable, it is essential to understand farm animals’ senses and the way they perceive their environment. Numerous studies have been undertaken on humans and the importance of a healthy working space. For example, exposure to high levels of noise has been linked to mental health [5]. Auditory pathways may act through nerves on the cardiovascular, gastrointestinal, endocrine, central nervous, and immune systems. The effect of noise is believed to occur through a network of nerves. These pathways, in conjunction with the autonomic nervous system, influence various systems in the body, including the cardiovascular, gastrointestinal, endocrine, central nervous, and immune systems. The connections between the auditory pathways and the reticular formation, a part of the brain, may account for the effects of noise on behavioral activation (arousal), pain, and sleep. Additionally, alterations in chemical signaling could contribute to certain neurobehavioral symptoms [6]. Still, there is a lack of understanding about these processes in animals and how the living environment (including sounds and noise) influences their welfare.

Considering the significant level of animal well-being, it is important to adapt living conditions according to the specific needs of each species. To do so, fundamental knowledge about their needs and perceptions is required. For example, inappropriate horse management may result in sleep deprivation, causing collapses and injuries [7].

Senses are systems by which animals receive information about their environment. Hearing, vision, olfaction, taste, and touch are the most common sensory modalities of vertebrates [8], whereas perception reflects how this information is organized, interpreted, and experienced. It can be explained by two processes: a bottom-up (physical) and a top-down (psychological) (Figure 1). Bottom-up processing reflects the transfer of the information from sensory input, whereas top-down processing refers to the way the information is interpreted based on the knowledge and experience gained through the lifetime [8,9].

Understanding how senses and perception work and what it means for animals’ behavior and physiology is essential for both science and practice. Probably the most studied sense is vision, and these findings have important practical outcomes (e.g., in horse show jumping, safety has been enhanced by replacing orange markers with bright yellow or blue ones) [10].

Knowledge about hearing is still limited and requires more attention in most species, yet it is far ahead of olfactory and tactile sensitivity cognizance. The sense of hearing, in both domesticated animals and their wild ancestors, plays a significant role in their lives [11]. The influence of noise has been recognized and included in minimum standards for keeping animals, e.g., Council Directive 2008/120/EC of 18 December 2008, which sets the minimum standards for the protection of pigs, states that no animal should be exposed to constant or sudden noise, and the noise level must not exceed 85 dB [12]. In fact, sudden sources of noise that could scare the animals (e.g., cattle [13], or pigs [14]) should be avoided for all species. 

It is important to acknowledge the role of hearing and sound to implement “the above five domain approach for animal welfare: a life worth living” [4]. Comprehension and proper interpretation of auditory functions are fundamental to animal–human and animal–animal interactions, especially in artificial conditions to which livestock is exposed.

The rearing environment of modern farm animals is poor in stimuli compared to their natural environment, which may lead to atypical behaviors [2]. This could be caused by a reduced level of natural sounds or too many artificially loud tones [11]. It is important to recognize the sources of sounds in the surroundings, which are the animals themselves, mechanical and electrical devices and equipment in the buildings and facilities, as well as people who operate them, but also some from the neighborhood (e.g., roads, rail tracks, etc.).

The aim of this study is to gather and sum up the current knowledge about the role of sound and hearing in farm animals. Information about senses and perception will be presented in relation to livestock’s natural needs and welfare and put into a practical context, outlining gaps for future research.

## 2. Definitions

A sound is an auditory impression caused by acoustic waves propagating in an elastic medium (solid, liquid, or gas). In the physical sense, sounds are not only “auditory effects” produced by musical instruments or vocalizations, but also all types of murmurs, noises, creaks, thumps, cracks, etc. Sounds are characterized by pitch, duration, loudness (volume), timbre, sonic texture, and spatial location [15]. Volume is expressed in decibels (dB), and pitch is determined in Hz. It is important to note that different species show sensitivity to sounds depending on their hearing organ’s anatomical structure. Moreover, their natural vocalization and the environment in which they live are also important in terms of adaptation and habituation. This sensitivity is the so-called hearing range (Table 1)—for example, dogs’ hearing range is 67–45 kHz (45,000 Hz) [16], while humans hear sounds with frequencies from about 20 Hz to about 20 kHz [17]. Acoustic vibrations with frequencies below 20 Hz, inaudible by an average healthy human, are known as infrasound (used, for example, by whales), and frequencies above 20 kHz, in turn, are referred to as ultrasounds [18] (used, for example, in ultrasonic dog whistles). It is crucial to consider animals’ abilities to hear sounds above and/or below the human hearing range, which is significant to their wellbeing and the safety of both humans and animals, as they may react to stimuli undetectable by humans (e.g., horses may exhibit a strong flight response, posing a risk to the rider). 

The extremes of the audible spectrum can be aversive, so people should be particularly attentive to these sounds [11].

Noise is difficult to define as it may depend on individual perception. The Cambridge Dictionary defines it as a sound that is unwanted, unpleasant, or loud [23]. Noise is a subjective quantity, and noise pollution is decided by the affected person [24]. From a physics standpoint, noise is a disordered sound of continuous frequency that is usually random. Environmental noise is a cumulation of sounds from the ambient environment and can be considered a pollutant that leads to annoyance and can be perceived as stressful [25].

## 3. Hearing Abilities of Farm Animals

The first visible reaction that can be acknowledged as proof of hearing is animals’ attention (head turn or ears turn) toward the source of sound. This reaction may be known as the pryer reflex. Hearing, in general, is a crucial sense for spatial orientation, identification of herd members, and recognition of dangers. 

Heffner and Heffner [21,22,26] studied farm animals’ hearing abilities in the 1980s, and this is the most common source of knowledge about frequency and loudness to which animals are sensitive (see Table 1). Most farm animals, except birds, are able to hear sounds at higher frequencies than humans. It is crucial to be aware of this, as high-pitched tones may be unpleasant. These sounds may be produced by some electronic devices, e.g., computers or broken lamps, or they may be used intentionally to repel rodents. 

It has been found that chickens are the most sensitive to sounds of 2.6 dB at 2 kHz and that domestic chickens hear much lower frequencies than humans: some studies demonstrate that their hearing range is 60–12 kHz [19], while other studies indicate that chickens can hear much lower frequencies ranging from 2 Hz to 9000 Hz [27]. However, as they require more training to respond to sounds below 64 Hz compared to higher frequencies, it is suggested that they perceive infrasound at a different quality. Birds’ anatomy is unlike mammals, with their ears located on both sides of the head with an external hole surrounded by special feathers that protect the entrance to the ear canal but do not block the transmission of sound [28]. Impressively, the development of hearing emerges already around the 12th day of incubation [29], which sheds new light on handling laid eggs. In mammals, what people usually call an ear is actually a pinna (a part of the outer ear). It is shaped to capture sounds from the environment and works as a funnel for acoustic waves. Horses, for example, have ten auricular muscles that allow the pinna to rotate 180 degrees, allowing them to direct attention to the sound without head movement, compared to the three muscles of the human ear [30].

Sheep have similar hearing sensitivity to humans, with a threshold of around 10 decibels (dB). However, their hearing range is shifted to higher frequencies, reaching up to approximately 42 kHz. This higher range potentially enables them to hear a dog whistle [20]. It would be interesting to learn more about the capabilities of sheep to make associations between the whistle and their behavior. Furthermore, their ability to perceive higher frequencies compared to humans may also expose them to potentially stressful effects of ultrasounds, such as those produced by machines, which are inaudible to humans.

The range of audible frequencies for goats is wider than that of sheep, as they are capable of hearing lower frequencies (as shown in Table 1). While goats exhibit relatively good sound localization precision (the capacity to determine the direction and distance of a detected sound expressed in degrees) among herbivores (around 18°), it is still considerably lower compared to humans or predators, such as cats, which can localize sounds with an accuracy of approximately 5.7° [26]. Additionally, goats are known for their selective hearing, which enables them to accurately identify other animals and/or people in their surroundings.

In general, the mass of a mammal is inversely proportional to its communication frequency [31]. However, horses are an exception to this rule, as they are large animals with limited ability to hear low frequencies. Horses are most sensitive to sounds in the range of 1 to 15 kHz [32]. Like most farm animals, they are also capable of hearing higher frequencies than humans. Their unique ability to move their ears in different directions allows them to orient their ears towards two different sound sources [33].

Sudden loud noises may scare horses, causing an immediate, rapid reaction that can endanger the safety of both people and the horses themselves [8].

In comparison to horses, pigs have larger ears that are also movable. The auricles of pigs are mobile, allowing them to better detect and locate sounds [34]. The anatomical structure of pig auricles can vary depending on the breed, which may also impact their ear movement [35]. Generally, larger and heavier ears are less mobile compared to smaller ones [36]. Pigs seem to have the best sound localization acuity among farm animals (4.6°) [21]. Moreover, they are adapted to hear better at higher frequencies than humans. Pigs react to sounds from 42 Hz to 40.5 kHz, with a region of best sensitivity from 250 Hz to 16 kHz [21]. Research [37] has shown that piglets exhibit stronger reactions, measured through behavior and heart rate, when exposed to sounds with higher frequencies and intensity. Noise, particularly high-volume sounds, can lead to aggressive behavior and weakened immunity in pigs [38].

Cows have a hearing range that is closest to that of humans, as shown in Table 1 [22]. However, they have a better ability to perceive higher frequencies. In contrast to horses and goats, cattle have a poor ability to localize the source of sounds [39].

In terms of farm animal welfare, it is crucial to consider the impact of high-frequency sounds that are inaudible to humans. Additionally, high-pitched tones may stimulate or even stress animals, while low-frequency sounds have a calming effect, such as in cattle [39] and horses [40]. Therefore, it is important to emphasize to employees the significance of staying calm and using low, soothing tones when interacting with animals. Proper use of sound and touch can alleviate stress and help animals adapt to barn conditions [41].

## 4. Vocalizations in Livestock Farming

Most animals primarily rely on body language for communication, but vocalization plays a role in the communication of farm animals as well. The types of sounds produced by animals are specific to their species, breed, sex, age, and the overall situation in which they find themselves. Animals use specific sounds to respond to various events in their environment and to express different emotional states. These sounds convey important information, such as warning of danger [42], mate selection [43], or calling for the feeding of offspring [44]. Farmers can benefit from paying attention to the sounds made by their livestock, as they can provide useful information.

Vocalization plays a special role in communication between a mother and her offspring, but even more fascinating is the situation where chicks use acoustic communication to synchronize hatching [45]. 

Studies on vocalization in pigs have been conducted for over half a century and have made significant advancements compared to other farm animals. Researchers have identified up to twenty-three different types of pig calls and even translated recorded vocalizations into musical notation [46]. In the 1970s, sonographic methods were introduced to analyze vocalizations in both wild and domesticated animals. Researchers have identified 14 distinct swine sounds, each linked to specific situations [47]. Ongoing discussions exist regarding whether vocalization is the primary or secondary mode of communication for pigs. It is hypothesized that pig vocalizations are strongly associated with their level of excitement. Pigs utilize their hearing not only to detect environmental sounds but also to communicate with each other. For example, short, single snorts are common during exploration; long series of grunts are used for greetings and movement; and high-pitched squealing is associated with pain or distress [48]. 

Vocalization between a mother and her offspring is extremely important in pigs. Sows communicate their readiness to feed using different sounds compared to calling for piglets in other situations. Sows use grunts to encourage piglets to suckle; when they start to nurse, the frequency of grunts rises, and finally they become quiet at the end of feeding [49]. During the initial massage of the sow’s udders and when competing with siblings for a spot at a desired nipple, piglets emit loud and intense squealing sounds. However, once they start to suckle milk, they become quiet. If piglets are in distress, they will vocalize to call for their parents. If they are being crushed by the sow or handled for various procedures such as weighing, tail and teeth clipping, vaccination, or blood collection, they will emit a long, high-pitched squeal. After that, they often rush over to the sow while making grunting noises, which can be interpreted as a form of communication indicating their discomfort or pain experienced during the prior event [50].

At the same time, piglets and their mothers utilize various forms of communication. Vocalization is a crucial method of communication between sows and their offspring [51] and can help reduce the number of piglets lost due to crushing (overlying) [52]. Piglets emit grunts or snorts during ordinary interactions or play sessions with their littermates. However, if these activities escalate into scuffles or fights, intense squealing can be observed. In social interactions among adult swine, short and intense squealing, combined with body language, signals the position of each animal within the group. Different vocalizations, typically low-toned, are used for social communication, while alarm and feedback signals are generally longer and louder, recognized as high-frequency vocalizations audible to humans as squealing [49,53]. These sounds may indicate a low level of welfare in cases where animals live in unnatural conditions [54]. Additionally, such sounds are produced by piglets in highly stressful situations, indicating a reduced level of well-being, such as during castration [22] and after weaning [23]. Proper interpretation of vocalizations, especially high-frequency sounds, can significantly contribute to understanding the underlying reasons for deterioration in swine welfare.

The vocalizations of farm animals serve as an excellent source of knowledge about their emotions, both positive and negative. Emotions experienced by animals lead to changes in the autonomic and somatic nervous systems, resulting in muscle tension, vocalization, breathing changes, and drooling [55]. In recent years, there has been increased attention paid to the emotions experienced by animals in response to certain events, driven by growing awareness and sensitivity towards animal well-being [56]. Domestic pigs heavily rely on vocal communication, and the sounds they produce vary depending on the situation they are in [57]. The variation in pig vocalizations could be attributed to their emotional state, specifically their level of positivity or negativity. However, the relationship between emotions and vocal expression in pigs is complex, as they use multiple types of calls in different contexts, and the acoustic parameters of each call type may change in various ways based on their emotional state [58]. Piglet sounds can be classified into two groups based on pitch: high-frequency (HF) and low-frequency (LF) calls, each with their own subcategories [57]. High-pitched HF calls, such as screams and squeals, are frequently heard in negative circumstances, while low-pitched LF calls, such as grunts, are more prevalent in neutral and positive situations [59]. Consequently, HF calls may indicate negative emotions. Nonetheless, there is considerable variability within each call type, including differences in length, tone quality, and energy distribution, which could provide further insights into the pig’s emotional state and the conditions under which the sound was produced. Vocalization can, therefore, serve as an indicator of the level of animal well-being in animal farming. Automated methods for call classification in pigs were tested by Briefer et al. [58]. Their results showed that positive calls were shorter. They also found that changes in duration and amplitude modulation rate may be useful for further development of automated systems aimed at recognizing emotional valence [58]. 

Specific sounds are made by animals before feeding time. Pigs demand their feed with a loud, intense squeal, which becomes even more pronounced when feeding staff appear or when there is a delay in feeding. Once the feed is dispensed into the trough, vocalizations are silenced. However, any scuffle at the trough results in long-lasting, intense squealing produced by the participating individuals [60].

Vocalization can be heard among all farm animals, although studies on this are very limited. As mentioned earlier, for pigs, vocal communication between mothers and offspring is a characteristic form of communication present in all species. In cows, sound communication between the cow and the calf affects the amount of milk produced [49]. It has been observed that cows produce different vocalizations when they face social isolation, when they are reunited with the herd after isolation, and when they want to maintain contact within the herd by using an ‘mm’ vocalization [47]. 

The most distinct sounds made by animals are during the mating season, known as rut or rutting, which can be observed in wild boars, swine, fallow deer, deer, elk, and foxes, as well as in most mammals. These sounds are believed to attract individuals of the opposite sex and indicate readiness for reproductive activities.

Studies conducted by Shillito [61] have shown that ewes can recognize the bleating of their lambs shortly after birth. It has been found that sheep can identify individual members of the flock as well as individual humans based on the sounds they make. Experiments using Y-maze tests have demonstrated that sheep can differentiate between ovine and human vocalizations [62]. When sheep want to express different emotional states, they produce various sounds, although these differences may not always be recognizable to people.

In goats, two main types of bleats can be recognized: open-mouth bleats and closed-mouth bleats [63]. Lower sounds are associated with more positive emotions [64]. Goats also have a strong “sound memory”; research has shown that a mother never forgets the voice of her offspring [65,66,67]. Less specific sounds produced by goats include various types of gurgles and snorts [68]. Research on goat vocalization is still limited.

Horses are often considered silent animals, but they can still produce different vocalizations and sounds in their communication, such as squeals, whinnies, and snorts [69]. Squeals are usually heard during agonistic interactions. Whinnies, which can take different forms from soft greetings (nickers) to high-pitched tones (whinnies), are produced during social isolation or when reunited [69]. There is a distinction between whinnies in negative or positive situations [70]. Like in most species, sounds associated with positive emotions have a lower frequency. Two different sounds produced by the nostrils can be distinguished: snorting and blowing. Snorts are pulsed sounds usually present in positive situations, while blowing is a non-pulsed, intense sound produced during stressful situations [71].

In general, the vocalization of farm animals can provide valuable information about their emotional state (please see review [3]).

## 5. Noise in Livestock Facilities

Facilities and buildings designed for livestock are filled with various sounds, often referred to as noises, which are typically produced by mechanical devices used in livestock management. These devices include feeders, fodder machines, troughs, watering systems, and feed lines to ensure microclimatic conditions (fans, heaters, automatic curtains), cleaning devices, and all sorts of vehicles.

The risk of chronic stress in animals due to prolonged exposure to noise is often overlooked. Long-term exposure to noise may lead to welfare deterioration, behavioral disorders, or the development of anxiety and fear. Noise can cause changes in physiological parameters (acceleration of heart rate, elevation of cortisol levels—[28,29,30], leading to weakened immunity and pathological changes, and consequently reducing productivity (lactation, fattening, delayed oestrus, etc.).

The level of stress is influenced by the volume of audio signals. Studies in poultry have shown that high noise volumes increase lymphocyte levels and the duration of tonic immobility in chicks. It has been noted that loud noises were associated with an increase in chickens’ corticosterone levels [72], suggesting elevated stress.

Given a choice, animals usually prefer tranquil places. For example, in Y-maze choice tests, heifers selected the maze arms that were quieter [73]. Recent studies in rats have shown that high-intensity noise causes oxidative stress, which can lead to damage in the organism [74]. Furthermore, it has been observed that noise can increase dopamine, adrenaline, noradrenaline [75], and corticosterone levels, leading to reduced humoral immunity [76].

It should be noted that even well-functioning devices produce noise (humming), which could potentially further decrease animals’ comfort. Prolonged exposure to humming sounds can result in chronic stress. A Swiss study [77] demonstrated that fan noise exceeding 85 dB extended the duration of pig fattening by 14 days and disrupted sow–piglet relationships during piglet rearing. Louder noise (95–110 dB) caused anxiety and an increased heart rate. Sudden sounds can also lead to disturbances during farrowing and an increased number of stillborn piglets [78]. 

The adverse impact of noise is recognized in legal provisions governing animal farming, such as Council Directive 2008/120/EC [12]. While complete elimination of noise may not be feasible, especially in large-scale livestock farming, efforts should be made to minimize and, whenever possible, eliminate noise due to our awareness of its detrimental effects. The reaction of individual animals to sound stimuli, referred to as noises, can be highly differentiated and depend on species, life experience, health status, etc.

## 6. Sounds Generated by Humans

The well-being of animals, often reflected in their behavior and productivity, can also be influenced by sounds produced by farm staff, such as loud conversations, calls, shouts, whistling, singing, and radio playing (including various types of music), in order to make work more enjoyable. Animals associate repeated sounds with specific activities, like cleaning pens, feeding, management, and veterinary procedures. Such sounds also make it easier for animals to identify particular individuals performing work around them [79,80].

Through classical conditioning, animals may develop associations with radio sounds, both positive and negative. Specific types of music or songs associated with particular individuals or categories of activities can have consequences. Such associations, which often arise spontaneously, can also be deliberately created by people to achieve specific goals, as demonstrated by the numerous successful attempts described in the following paragraphs.

Young horses exposed to music with a steady rhythm demonstrated a lower heart rate, calmer behavior, and spent more time foraging [81]. In another study, it was observed that the type of music may influence horses. The trend of using country music to increase the time spent eating was observed, while jazz had the opposite effect. Nevertheless, it is important to note that this research was conducted with small groups and did not yield statistically significant results [82]. Music played to purebred Arab horses before their sport training had a positive impact on their welfare and race performance [83]. 

The positive effect of classical/jazz music on farm animals’ welfare has been suggested, but more data are still required [84,85].

These associations may have practical applications for reducing stress. For example, the music was played for young piglets every day for 15 min, while they were allowed to leave the pen. This created an association between the music being played and a pleasant experience—a chance to move around in a larger space. Piglets in the control group listened to the same song without having the ability to leave their pen. After weaning, the same song was played, but the animals of both groups remained in the pen. The piglets in the experimental group exhibited greater mobility and less aggression, resulting in better health and larger weight gains [86]. In another study, music was successfully introduced as a potential environmental enrichment for pigs, and different emotional responses were observed depending on the harmonic characteristics of the music [87].

Furthermore, cows’ adaptation to an automatic milking system was enhanced by music being played during milking [88].

Sound signals are often used in animal training. Voice cues or whistle cues are used to indicate desired behavior. In clicker training techniques, a bridge stimulus (such as a clicker, voice, or whistle) informs animals when they have done something correctly. The sounds used can be voice commands, whistling, or noises made, for example, by a clicker [89]. The previously used training methods based on punishment are being increasingly replaced by methods that utilize positive reinforcement and motivate animals to display desired behaviors [90].

The impact of sound stimuli on the well-being of humans is well-known and relatively well-documented. Music is used to reduce pain and combat depression, stress, and anxiety [91]. Auditory stimuli received by listeners can affect their mood and behavior. Music can also affect cognitive abilities [92] in both animals and humans [51]. This knowledge, along with information about the diverse reactions of animals to various types of sounds, including different music types, encourages researchers to undertake more insightful studies focusing on both cognitive and practical aspects.

## 7. Conclusions

There is a wide range of research papers available on the negative impact of noise on animal welfare, but most of the studies focus on rats rather than livestock. More research is needed in real farm environments to better understand the existing challenges. Fortunately, there are an increasing number of papers exploring the emotional states linked to vocalization and their relationship to animal welfare, which is a significant step towards gaining a deeper understanding of animals. However, there is still insufficient research on the practical implementation of sounds and music in agriculture. The papers cited here suggest the potential use of music in reducing stress, but further attention should be given to connecting scientific evidence with practical applications, especially regarding noise management on farms. In order to bring about changes in animal management practices, strong evidence is necessary to identify each factor that causes a reduction in farm animal welfare.

In the diagrams given in Figure 2, Figure 3, Figure 4, Figure 5 and Figure 6, the most important effects of sound and noise on different species are presented.

## Figures and Tables

**Figure 1 animals-13-02307-f001:**
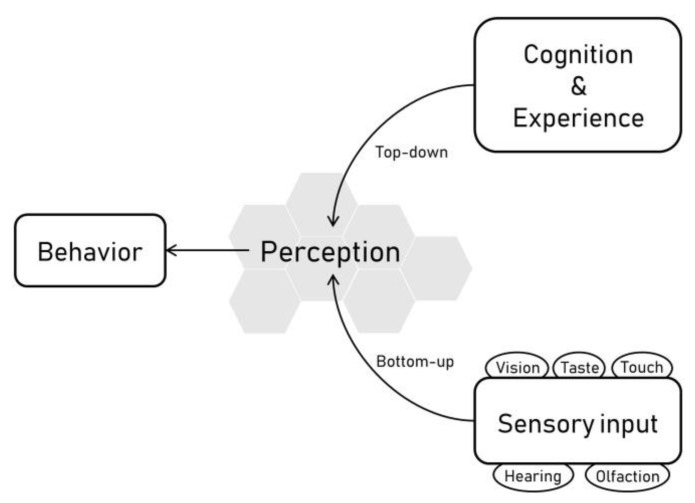
Perception and its bottom-up and top-down processes. Behavior is the result of both [8].

**Figure 2 animals-13-02307-f002:**
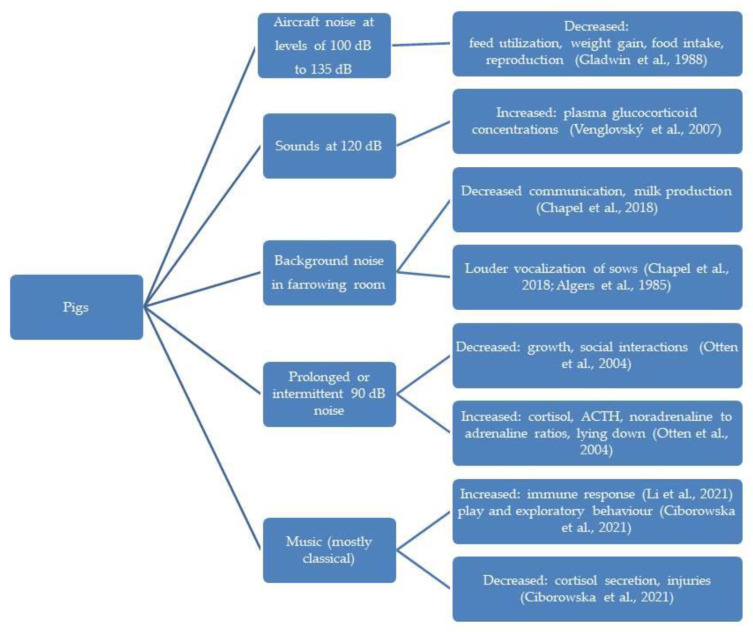
Effect of sound and noise on pigs [84,93,94,95,96,97,98].

**Figure 3 animals-13-02307-f003:**
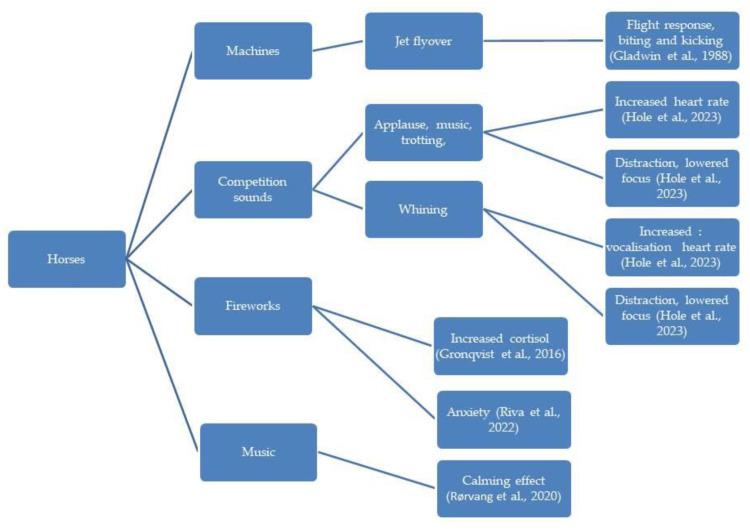
Effect of sound and noise on horses [8,93,99,100,101,102].

**Figure 4 animals-13-02307-f004:**
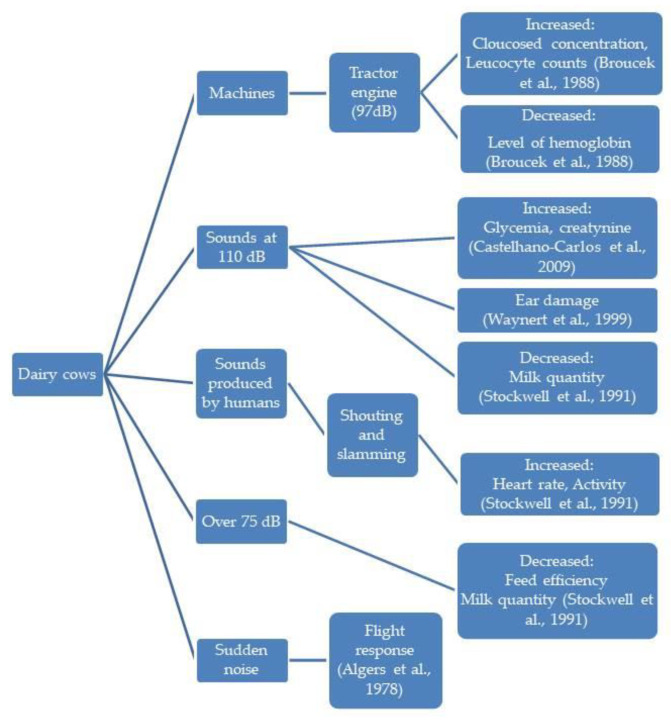
Effect of sound and noise on cows [103,104,105,106,107].

**Figure 5 animals-13-02307-f005:**
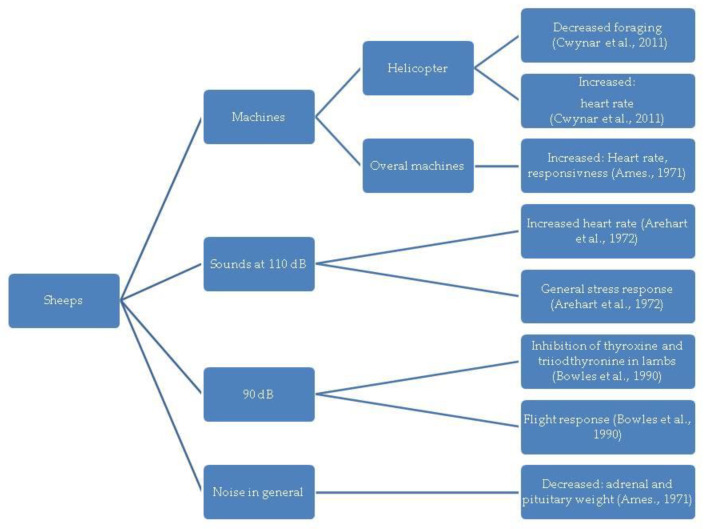
Effect of sound and noise on sheep [108,109,110,111].

**Figure 6 animals-13-02307-f006:**
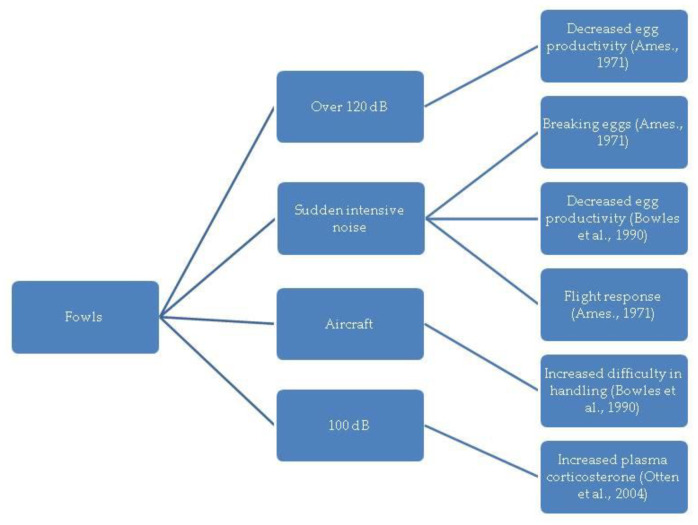
Effect of sound and noise on fowl [97,109,111].

**Table 1 animals-13-02307-t001:** The hearing range of farm animals and humans.

Species	Hearing Range (kHz)	Reference
Chicken	60–12 kHz	[19]
Sheep	125–42 kHz	[20]
Goat	60–40 kHz	[21]
Pig	42–40.5 kHz	[21]
Horse	55–33.5 kHz	[22]
Cattle	23–35 kHz	[22]
Human	20(40)–20 kHz	[17]

## Data Availability

Not applicable.

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
