# Peer review of "The Role of Sound in Livestock Farming—Selected Aspects"

_animals, 2023, doi:10.3390/ani13142307_

Round 1

Reviewer 1 Report

Review

The manuscript, “Role of sound in livestock farming - selected aspects” by Olczak et al. aimed to gather and sum up the current knowledge about role of sound and hearing in farm animals. Sounds on farms may come from machines, humans, or animals themselves. Vocalization may be very informative to the breeder as it is an expression of an emotional state. Taking into account that information can be very helpful in upholding a high level of livestock welfare. Furthermore, an understanding of learning theory, conditioning, and how some sounds may be beneficial to animals could be implemented in farm management to decrease the aversiveness of some stressful events.

Comments and Suggestions for Authors:

1- Please correct the keywords according to MeSH.

2- Every succeeding paragraph should be indented.

3- In line 48, the phrase "see the review [6]" is given. Please include the parts you want from this review in this article and refer to the review article.

4-Figure 1 does not have the necessary quality, and there is no need to write the word "Source" in its description, and you can simply write the appropriate reference.

5-Why lines 83 to 88 are not referenced?

6-Please specify in a diagram the effects of sound on different animals.

7-Determine which sounds with which frequencies have negative effects on animals.

.

Author Response

Thank you for your support and valuable comments.

  1. Please correct the keywords according to MeSH.
    Thank you for your suggestion. Keywords were updated according  to MeSH.

  2. Every succeeding paragraph should be indented.
    The suggestion was included. The text has been corrected. We strongly apologies for some editorial mistakes, we took effort to correct everything.

  3. In line 48, the phrase "see the review [6]" is given. Please include the parts you want from this review in this article and refer to the review article.
    Thank you for pointing that out. We did correct that part and managed to add most important information from the article.

  4. Figure 1 does not have the necessary quality, and there is no need to write the word "Source" in its description, and you can simply write the appropriate reference.
    The image has been replaced and enlarged. The word source has been removed.

  5. Why lines 83 to 88 are not referenced?
    We appreciate your suggestion to our work. Two references has been added.

  6. Please specify in a diagram the effects of sound on different animals.
    We couldn’t have done it better without your suggestion. We believe adding some diagrams improved the paper.

  7. Determine which sounds with which frequencies have negative effects on animals.
    Some clarification has been added under the table with hearing range.

Reviewer 2 Report

Line 40 – David Mellor is arguing that the five freedoms is not sufficient for good welfare, rather he suggests the five domain model is important as it includes mental well-being.

Although this is an interesting area ad the authors are clearly knowledgeable, there were several sections where the findings from papers were mainly listed rather than fully discussed. Much more indepth discussion would have been useful.

The structure could also be reconsidered - there was some unnecessary repetition. E.g., separating natural vocalisations from noise in facilities and soubds generated by humans meant that the effects were sometimes repeated. I wonder whether a section early on about the negative effects of noise would have been better and then go on to consideration of specific research on the various noise situations. Then any positive effects of noise.

There are too many instances of typos and grammatical mistakes to be listed here. I cannot imagine how difficult it would be to write an academic paper in a second language. The authors should be commended for this. However, I recommend that you access someone who has English as their first language to provide the extensive corrections needed.

Author Response

Line 40 – David Mellor is arguing that the five freedoms is not sufficient for good welfare, rather he suggests the five domain model is important as it includes mental well-being.

Thank you for important note. The sentence at line 40 has been reworded.

 Although this is an interesting area ad the authors are clearly knowledgeable, there were several sections where the findings from papers were mainly listed rather than fully discussed. Much more indepth discussion would have been useful.

Thank you for this important note. It is right that deep discussions are really needed in terms of animal welfare. However, we aimed to  gather information about different farm animals in one place. We limited paper to point out most important issues connected to sounds and hearing. Deep discussion of each aspect would require the whole new paper for each species separately.

The structure could also be reconsidered - there was some unnecessary repetition. E.g., separating natural vocalisations from noise in facilities and soubds generated by humans meant that the effects were sometimes repeated. I wonder whether a section early on about the negative effects of noise would have been better and then go on to consideration of specific research on the various noise situations. Then any positive effects of noise.

We would like to acknowledge the work and thinking through that has been done. We hope that maybe instead of changing  the structure, added diagrams will be in line with your suggestions. We have  added diagrams  to sum up most important factors in one place with visible and easy to read way. While doing so, we have extended a paper by some important information.

The paper has been corrected by native speaker.

Round 2

Reviewer 1 Report

Correctins are fine.

Author Response

Thank you.

Reviewer 2 Report

Thank you for responding to my feedback. I agree that the additional diagrams are very useful.

The paper is much improved but there are still many errors of expression.  Please see the attached scanned version of your paper with errors highlighted and corrections suggested.

I have included suggested corrections in the attached document (above).

Author Response

Thank you for your feedback and detailed help to improve the paper.

All suggestion has been corrected.